# AprilFEC: Real-Time Channel Estimation and Adaptive Forward Error Correction

Ryan J. Marcotte, Xipeng Wang, and Edwin Olson

University of Michigan

{ryanjmar,xipengw,ebolson}@umich.edu

*Abstract*—Transporting large amounts of data over the wireless links of a mobile ad hoc network requires mitigating significant and unpredictable packet loss. Traditional acknowledgment-based transport protocols (e.g. TCP) perform poorly as ACK messages are dropped and the transport falls back on retransmission timeouts. We propose a novel application-layer transport system, AprilFEC, that employs erasure codes to ensure reliable delivery over lossy, time-varying wireless links. Moreover, AprilFEC minimizes added overhead by estimating packet loss rates and adapting its encoding level accordingly. We show that AprilFEC delivers large media files over lossy, time-varying links more reliably than a traditional TCP-based system.

## I. INTRODUCTION

While modern wireless links are capable of very high peak bandwidths, their empirical performance in a mobile robotics setting can vary over orders of magnitude. As nodes in a robotic network move through their environment, link qualities fluctuate and network performance suffers. Even low-bandwidth communication requirements can be hard to meet in high-loss networks; high-bandwidth communication becomes virtually impossible. We propose an end-to-end application-layer system, AprilFEC, that reliably transports data over variable-quality network links through the adaptive application of erasure coding, significantly improving performance for network-intensive tasks such as media streaming.

The traditional approach to reliable network transport is a scheme of acknowledgments (ACKs) and retransmissions. In such a protocol, the source node repeatedly transmits a packet until the destination node acknowledges its receipt by replying with an ACK message. ACK-based protocols pervade modern networking stacks because they are simple and largely effective across a wide range of network types and applications. However, these protocols have primarily been developed for use in wired and wireless networks with fixed physical infrastructure and low packet loss. Mobile ad hoc networks (MANETs), such as those used by many robotic systems, lack such infrastructure and can exhibit high packet loss, causing ACK-based protocols to perform poorly as ACK messages are regularly dropped and the transport falls back on retransmission timeouts. For example, TCP displays significant performance deterioration when loss rates exceed 10 percent [15]. Mobile robotic networks often experience packet loss events well in excess of this level [8], requiring an alternative approach to reliable delivery.

In this paper, we propose AprilFEC, a reliable application-layer transport mechanism suited for the lossy, time-varying

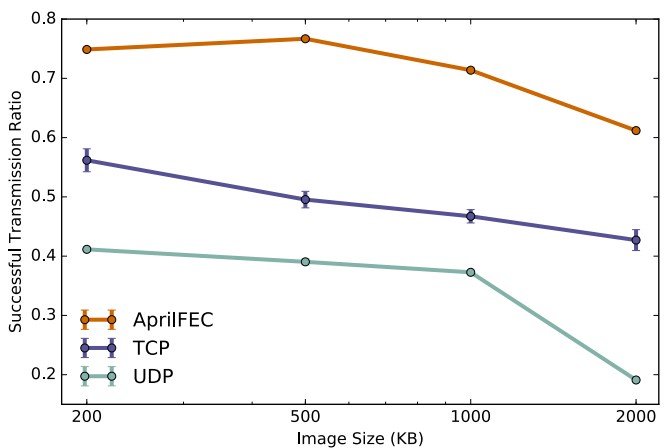

Fig. 1. Proportion of successful transmissions during image streaming simulations driven by real-world robotic network data. AprilFEC significantly outperforms TCP, the traditional reliable-delivery protocol. When packet loss rates are high, ACK messages are frequently dropped causing TCP to stall on retransmission timeouts. By comparison, AprilFEC uses an adaptive forward error correction scheme to deliver data with high likelihood even under poor network conditions.

links of a robotic network. AprilFEC is built on top of UDP and employs erasure codes to recover data lost during transmission. More specifically, AprilFEC adaptively applies fountain codes [6], a type of erasure code that efficiently produces variable levels of redundancy (see Sec. III). In this way, AprilFEC supports reliable message delivery while minimizing the overhead it incurs on the network. We show in Fig. 1 that AprilFEC significantly outperforms TCP at the demanding task of streaming high-resolution imagery over a time-variable, lossy wireless link.

Our specific contributions include:

- A kernel density estimation technique for characterizing the instantaneous packet loss rate of a channel (Sec. IV-B),
- An inference mechanism to predict the amount of forward error correction needed to achieve a given level of confidence for successful delivery (Sec. IV-C), and
- Robust and repeatable performance evaluation with a network emulator driven by packet traces from a real-world robotic network (Sec. V).

## II. Related Work

This paper builds upon many of the ideas previously presented in [8]. In that paper, we propose a system that adaptively applies forward error correction (FEC) to decrease effective packet loss rates even as raw loss rates grow. The primary contribution of the previous work is the identification of a tradeoff between latency tolerance and the reliability of delivery in such a system. Namely, when the transmission of encoded data has a wide temporal distribution, the system is more reliable in the face of burst errors typical to wireless networks.

This finding forms the basis for AprilFEC, but we make several distinguishing contributions in this paper. AprilFEC's kernel density estimator is better suited to packet loss rate estimation than the single-hypothesis Kalman filter of the previous paper (see Sec. IV-B). In contrast to the Reed Solomon codes used in the previous paper, AprilFEC employs fountain codes, which allow for highly efficient adaptation of encoding strength and scale to very large input sizes, such as the image files we transmit in our evaluation (see Sec. III). Finally, we present a more repeatable evaluation consisting of simulations driven by real-world network traces (see Sec. V-A).

Winstein and Balakrishnan [15] propose Sprout, an adaptive transport system designed to improve the performance of time-varying cellular networks. Similar to AprilFEC, Sprout makes stochastic forecasts of link quality and adjusts its transmission strategies based on these estimates. In particular, Sprout varies the rate at which it transmits packets in order to reduce self-induced network congestion. The authors demonstrate that Sprout can reliably support network-intensive tasks such as video conferencing for mobile internet users.

Gu and Grossman [4] propose UDT, a high-performance data transfer protocol designed for use over high-speed wide area networks. Like AprilFEC, UDT is built on top of UDP. To make UDP more reliable for the transport of large quantities of data, UDT adds its own congestion control and selective acknowledgments. UDT sends ACKs at a fixed interval, so they consume very little overhead when data is transferred at high-speed. However, for low-bandwidth traffic, UDT acknowledges each packet individually, leading to poor performance over lossy links.

Google introduces QUIC (Quick UDP Internet Connections) [13] as a reliable UDP-based replacement for TCP. QUIC includes numerous enhancements such as decreased latency in establishing connections, better congestion control, and forward error correction. In contrast to AprilFEC, QUIC's FEC module does not estimate link quality or attempt to vary redundancy levels, instead sending a small fixed ratio of redundant data.

Mahajan et al. [7] present an FEC scheme that opportunistically exploits spare network capacity through the application of erasure codes. By estimating the instantaneous load on the local network, the system is able to add redundancy without hogging bandwidth needed for other transmissions. The authors evaluate this technique on network-connected vehicles and show significant improvement in throughput.

Much research has been devoted to improving the performance of TCP under the moderate amounts of packet loss that occur in traditional wireless networks. One of the best-known examples is TCP Westwood [9], a variant of TCP intended to improve throughput over wireless links. TCP Westwood accomplishes this by using information from the ACK stream to adapt the congestion control window parameters. Grieco and Mascolo [3] show that TCP Westwood can improve utilization of wireless links affected by losses not due to congestion. In a similar vein, Bai et al. [1] seek to improve the performance of TCP over lossy links by distinguishing losses that occur due to network congestion from those caused by data corruption. To make this differentiation, the authors borrow a branch prediction technique originally designed to mitigate control hazards in CPUs.

## III. Fountain Code Primer

Understanding AprilFEC requires some background in the fountain codes it is built upon. The papers of MacKay [6] and Qureshi et al. [11] have comprehensive treatments of fountain codes. In this section, we briefly provide the reader with an overview of fountain codes as well as introduce online codes [10], the specific fountain code variant employed by AprilFEC.

Erasure codes are a type of forward error correction in which data is fragmented into small chunks (*symbols*), and encoded to produce a new set of symbols containing redundancy information about the original data. If some of the symbols are lost during transmission or storage, the encoded symbols can be used to recover the original data.

Erasure codes are parameterized by the number of input data symbols $k$ and the number of encoded output symbols $n$. Fixed-rate erasure codes, the most well-known of which are Reed-Solomon codes [12], are designed to operate with a fixed code rate (i.e. fixed values of $k$ and $n$). Varying the size of the input or amount of redundancy requires the initialization of a new code instance, or *codec*. As a consequence, inreasing the code rate (e.g. to produce additional redundancy) for an already encoded input requires replacing the set of existing encoded symbols. A significant benefit of most fixed-rate erasure codes is their *optimality*, meaning that any subset of $k$ encoded symbols can be used to recover all $k$ of the input symbols.

By comparison, rateless (fountain) codes operate on a fixed input size but can efficiently produce an unlimited number of encoded symbols from that input. In other words, a given fountain codec has constant $k$ but variable $n$. Fountain codes are non-optimal, but the guaranteed recovery properties of fixed-rate codes are replaced with probabilistic guarantees that the $k$ input symbols can be recovered from $k'$ encoded symbols with high probability, for $k'$ slightly larger than $k$. As $k'$ increases, the likelihood of decoding failure decreases exponentially. Furthermore, fountain codes are *locally encodable*, meaning that each encoded symbol only depends on a constant-sized fraction of the input and not on any of the other encoded

symbols. In other words, the value of $n$ does not have to be fixed at the time of encoding, and additional encoded symbols can be computed incrementally.

Currently, most state-of-the-art fountain codes such as Raptor Codes [14] are covered by patents (e.g. [2]) that limit their utility for the robotics research community. In an effort to make AprilFEC useful to the community, we have selected a fountain code variant, online codes [10], that we believe is not patent-encumbered, though users should consult their own expert legal advice. AprilFEC's design depends very little on the specific underlying fountain code, and its performance could be improved if a more efficient fountain code were to become available.

AprilFEC could also incorporate a fixed-rate code such as Reed-Solomon for small data sizes and low packet loss rates. This would decrease overhead added by the system but would not scale to larger data sizes and high packet loss rates. In this paper, we only consider an AprilFEC implementation with fountain codes in order to transport large quantities of data.

## IV. APPROACH

A naive application of fountain codes to the problem of reliable transport might entail continually transmitting encoded fragments until an ACK is received indicating that the transmission should terminate. The analogy often referenced in the fountain code literature is that of a bucket (the receiver) placed under a fountain (the transmitter) that collects water droplets until it is full (the original message can be decoded), at which point the fountain can shut off. As we will show in our evaluation (see Sec. V), this naive fountain approach represents an upper bound in terms of delivering data but leads to prohibitive levels of overhead when applied to networks with delay and packet loss.

In this section, we present AprilFEC, which refines the naive fountain idea with packet loss estimates that inform its decision of how much redundant data to transmit. Though AprilFEC can use ACKs to improve efficiency, the channel estimate removes the system's dependence on them. The overall system improves reliability over lossy links with modest added overhead.

### A. System Overview

Once a user passes data into AprilFEC, the system splits the data into smaller fragments suitable for transmission in the payloads of UDP packets. Based on a conservative estimate of the instantaneous packet loss rate, AprilFEC computes and transmits a batch of encoded fragments. The system momentarily pauses and waits to receive an acknowledgment from the destination node, at which point it terminates the transmission process. If no ACK is received, AprilFEC repeats the process with a higher estimated packet loss rate. After several iterations, AprilFEC concludes the transmission process, even in the absence of any ACK. The AprilFEC transmission procedure is given in Algorithm 1.

At the destination node, AprilFEC collects encoded fragments and attempts to reconstruct the original message from

---

**Algorithm 1** AprilFEC Transmission Procedure

**function** APRILFECTRANSMIT(data, $f$, $C$)
    $k' \leftarrow (1 + \epsilon)(1 + 0.55\epsilon q)k$      $\triangleright$ e.g. $\epsilon = 0.1, q = 3$
    $n_0 \leftarrow 0$
    **for** $t = 1 \ldots T$ **do**
        $p_t \leftarrow F^{-1}(c_t)$
        $n_t \leftarrow k'/(1 - p_t)$
        **for** $i = n_{t-1} \ldots n_t$ **do**
            $z \leftarrow$ GENERATEFRAGMENT(data)
            UDPTRANSMIT($z$)
        **end for**
        WAIT($w_{max}$)
        **if** ACKRECEIVED **then return**
        **end if**
    **end for**
**end function**

---

them. Once it is able recover the original data, AprilFEC passes the data to the user and sends an ACK back to the transmitting node.

### B. Packet Loss Rate Estimation

AprilFEC's estimate of the instantaneous packet loss rate informs its decision on the amount of encoded fragments it should transmit. The primary input to this estimation process is a periodic link quality report fed back from the receiver to the transmitter at a fixed rate. This feedback process is based on sequence numbers contained within each transmitted fragment. For a particular reporting period, the receiver tracks the number of fragments it receives and estimates the number of fragments transmitted by observing the largest and smallest observed sequence numbers.

The packet loss rates reported by the receiver may vary dramatically between intervals, especially as the signal-to-noise ratio of the link diminishes and packet loss rates exceed nominal levels (see Fig. 2 for example measurements). Because of this, single-hypothesis estimates (e.g. the one-dimensional Kalman filter used in our previous paper [8]) have difficulty predicting the true loss rate.

We instead utilize kernel density estimation to produce a packet loss rate distribution. Specifically, we use a sliding window of $n$ observations as samples in the distribution

$$f(x) = \frac{1}{nh} \sum_{i}^{n} K(\frac{x_i - x}{h}), \tag{1}$$

where $K(u)$ is an Epanechnikov kernel given by

$$K(u) = \begin{cases} \frac{3}{4}(1 - u^2) & |u| \leq 1 \\ 0 & \text{o.w.} \end{cases}. \tag{2}$$

In the absence of loss rate observations, we periodically evolve the distribution $f$ through the application of Brownian motion.

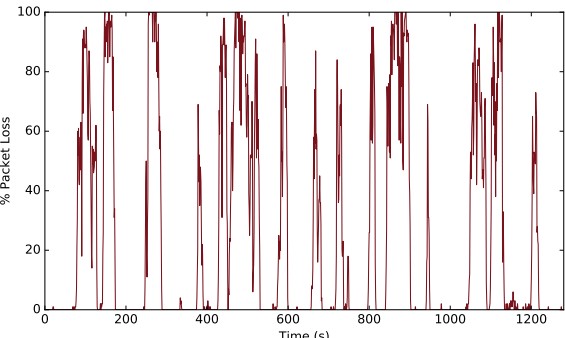 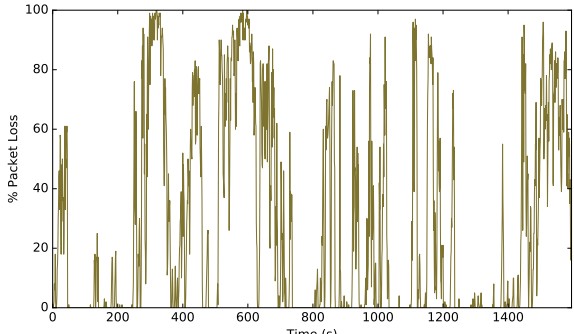

Fig. 2. Time-series data of packet loss rates taken from mobile robotic network traces in indoor (*left*) and outdoor (*right*) environments. Note the frequent and rapid fluctuations in loss rates as well as the outages and near outages that occur. In total, the two traces comprise nearly 50 minutes of real-world network data.

### C. Computing Redundancy Amounts

Given an estimate $p$ of the instantaneous packet loss rate, a decision must be made about the amount of encoded fragments to transmit. This decision depends not only on the loss rate distribution but also on the fountain code recovery properties.

Recall from the discussion of fountain codes in Sec. III that a message of $k$ symbols can be recovered with high probability from $k'$ encoded symbols for $k'$ slightly larger than $k$. The particular value of $k'$ and its recovery characteristics are properties of the specific fountain code in use.

In AprilFEC, we use the online codes of Maymounkov and Mazieres [10], which are parameterized by two values, $\epsilon$ and $q$. Conceptually, $\epsilon$ determines the degree of suboptimality of the code (i.e. the number of encoded symbols required for recovery) while $q$ affects the success probability of decoding. Both parameters affect the complexity of decoding. In our evaluation, we use the values $\epsilon = 0.1$ and $q = 3$.

As described by Maymounkov and Mazieres [10], an original message of $k$ symbols can be recovered from any $k'$ encoded symbols with probability $1 - (\frac{\epsilon}{2})^{q+1}$, where

$$k' = (1 + \epsilon)(1 + 0.55\epsilon q)k. \tag{3}$$

This value of $k'$ determines the number of encoded fragments that must be received in order for the data fragments to be recovered with high probability. For a link with estimated packet loss rate $p$, we transmit $n = k'/(1 - p)$ encoded fragments.

### D. Iterative Transmission

We combine the computations of the previous two sections to yield the iterative transmission procedure given by Algorithm 1. We begin the transmission process with the input data, the instantaneous packet loss distribution $f$, and a set of non-decreasing confidence interval values $C$, where $c_i \in (0, 1]$. The transmission procedure alternates between sending a set of encoded fragments and waiting for an ACK from the receiver. At each iteration $t$, the procedure computes the value of the quantile function $F^{-1}$ at the confidence level $c_t$ to yield an

estimated instantaneous loss rate $p_t$. That is,

$$p_t = F^{-1}(c_t) = \min\left(x \in [0, 1] : c_t \leq F(x)\right), \tag{4}$$

where $F$ is the CDF corresponding to the packet loss rate distribution $f$ defined in (1). The system uses $p_t$ to determine the number of encoded fragments that would need to be transmitted for a loss rate $p_t$. The procedure generates and transmits enough encoded fragments to account for the incremental increase in loss rate from $p_{t-1}$ to $p_t$. Finally, the procedure waits for time $w_{max}$ to determine whether an ACK is returned by the receiver. If an ACK is received, the transmission process terminates.

Note that this iterative transmission process is configurable depending on application demands. The set of confidence values can by modified according to a user's latency or overhead tolerance. In the case that no added latency can be tolerated, a single confidence value can be utilized, and AprilFEC will terminate transmission after one iteration without waiting for an ACK.

## V. EVALUATION

In the sections that follow, we evaluate AprilFEC on the task of continually delivering high-resolution image data over a lossy wireless link. We show that AprilFEC significantly outperforms TCP, successfully transporting data even as packet loss rates reach as high as 90 percent.

### A. Methodology Overview

As with many other areas of robotics research, designing realistic and repeatable evaluation procedures for networks remains a challenge. Simulators facilitate consistent measurement but may fail to adequately model the spatiotemporal complexities that affect a robotic network with mobile nodes. Real-world experiments, though valuable for validation, do not lend themselves to robust conclusions due to the cost of conducting a significant number of trials.

We borrow a hybrid technique from the networking community (e.g. [15]) to promote realistic and repeatable network evaluation in the field of robotics. Namely, we collect network condition data during real-world field robotics tests

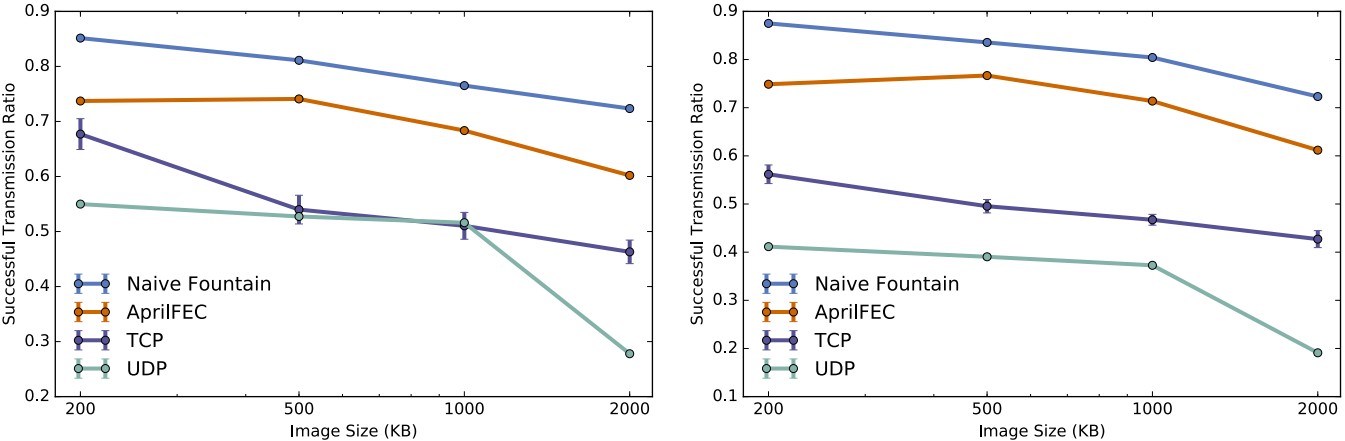

Fig. 3. Proportion of successful transmissions during trace-driven simulations in the indoor (*left*) and outdoor (*right*) environments. AprilFEC significantly outperforms TCP, the traditional reliable delivery protocol. Though a naive fountain approach has a higher proportion of successful transmissions, the overhead it incurs on the network is prohibitive (see Fig. 6).

and then use the resulting network traces to drive a local network emulator. Such an evaluation approach represents a reasonable compromise between the realism of field tests and the repeatability of simulations.

### B. Network Trace Collection

We collected network traces from a real-world mobile robotic system operating in indoor and outdoor environments. The trace collection runs consisted of a single point-to-point link between one mobile node and one stationary node (an "operator station"), both equipped with an OpenMesh OM2P-HS router. The routers run a stock version of OpenWRT 15.05, which implements IEEE 802.11s.

The network traces consist of the instantaneous packet loss rate and transmission bitrate measured on the link at a frequency of 1 Hz. To measure packet loss rates, we developed a custom tool that continually transmits UDP packets at a target bandwidth (e.g. 10 Mbps) and uses sequence numbers to track which packets are received or lost. We obtained the bitrate data by querying the operating system with netlink sockets.

We conducted two trace collection tests, one in an indoor office environment and one in an outdoor environment with various line-of-sight obstructions, including trees, hills, and small buildings. Both environments were slightly larger than the effective range of the routers, and the mobile node operated without consideration for the status of the network. As a result, the traces contain a wide range of network conditions, including some outages. In total, these traces comprise nearly 50 minutes of real-world network data. We visualize the packet loss rates of the network traces in Fig. 2.

### C. Trace-Driven Simulations

We used these network traces to drive simulations conducted locally on a single host. The trace data served as input to periodic calls to netem [5], a utility that manipulates Linux traffic control facilities to emulate various network

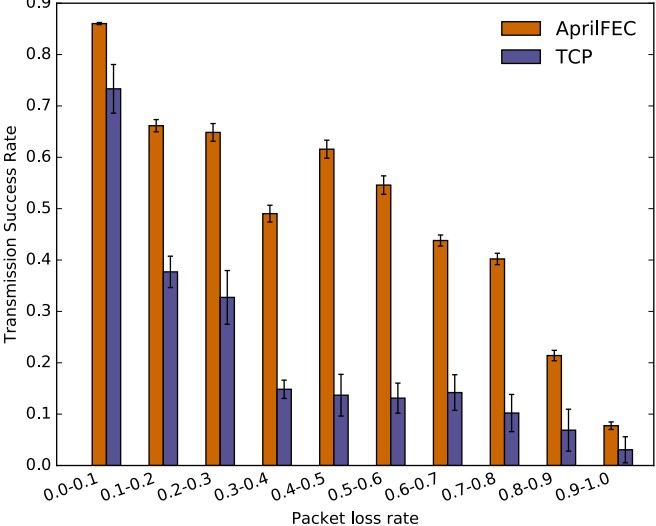

Fig. 4. Average proportion of successful transmissions of 1MB image on the indoor trace as a function of packet loss rate. We observe that even as loss rates increase, AprilFEC continues to deliver images, with its relative performance advantage over TCP being largest in high packet loss regimes.

conditions, such as delay, packet loss, or bitrate. For each tested protocol (UDP, TCP, AprilFEC, and the naive fountain disussed in Sec. IV), a transmitting application passed images of varying sizes across the local network. At a rate of 1Hz, netem modified network properties according to the statistics contained in the trace.

### D. Image Delivery Results

The primary metric we use to evaluate the performance of the various protocols is the average proportion of successful transmissions during each simulated trial. We tested each protocol on images ranging in size from 200KB to 2MB. For each protocol and parameter setting, we conducted 10

independent trials. Note that all error bars in the figures correspond to one standard error on the mean.

Fig. 3 shows the results of these experiments on the indoor and outdoor traces. In both experiments, the naive fountain approach serves as an upper bound on performance at each image size. This result is unsurprising since the naive fountain transmits as many encoded fragments as possible, making no attempt to minimize overhead. In the following section we will explore the prohibitive cost of the apparent high performance of the naive fountain.

As a more practical protocol, AprilFEC consistently outperforms TCP across all image sizes, averaging between 8 and 54 percent more successful transmissions per trial. Fig. 4 breaks these successful transmissions down by the ground-truth packet loss rate. We observe that even as loss rates climb to severe levels, AprilFEC continues to deliver images successfully. AprilFEC's advantage over TCP is most distinct at these high loss rates. For example, when the loss rate is between 40 and 50 percent, AprilFEC averages nearly 5 times more successful transmissions compared to TCP.

During some portions of the simulated trials, TCP actually performs worse than UDP, a consequence of the statefulness of TCP. A TCP connection that has stalled during a network outage may take time to recover even once link quality improves. In contrast, the stateless UDP is able to complete transmissions successfully as soon as packet loss rates drop back to 0.

### E. Overhead Results

For any protocol employing forward error correction there exists a tradeoff between the reliability of the protocol and the overhead it imposes on the network. We have designed AprilFEC to be a reasonable compromise between these two factors. Note that we measure overhead from the perspective of the receiver, that is, how many encoded fragments are received compared to the number of data fragments comprising the input message. There may be many more encoded fragments transmitted, but this type of overhead is the unavoidable cost of mitigating packet loss. We focus our attention here on overhead that is potentially avoidable, counting the number of encoded fragments received beyond the original number of data fragments.

In our evaluation, we explore two different types of overhead added by a fountain code-based system.

*1) Code Overhead:* The code overhead ratio is the ratio of the number of encoded fragments required for decoding, $\tilde{k}'$, to the number of original data fragments, $k$. This ratio is a function of the type of erasure code used and its parameters and is independent of the transport protocol. A value of 1.0 corresponds to an optimal erasure code such as Reed-Solomon, whereas all fountain codes have values greater than 1.0.

Fig. 5 shows a CDF of the code overhead ratio across all trials. We observe that the mean value of this distribution is 22 percent, with the $1^{st}$ and $99^{th}$ percentiles being 17 percent and 29 percent, respectively. These values could be improved

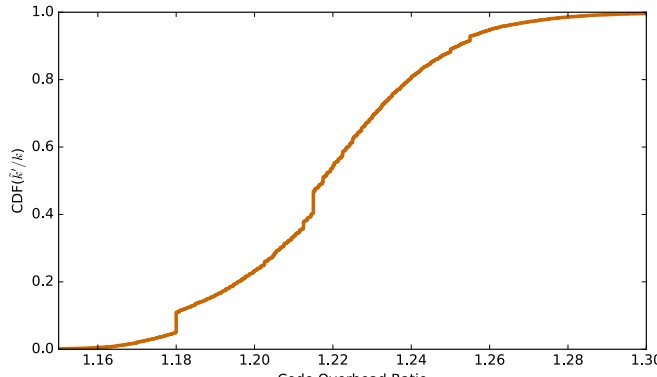

Fig. 5. CDF of the overhead added by the fountain codes themselves. For example, a code overhead ratio of 1.25 means that the decoding process succeeded once the number of encoded fragments received was 25 percent more than the number of data fragments. The overhead level is a function of the particular fountain code variant (online codes [10] in this case) and could be improved if a more efficient code was available.

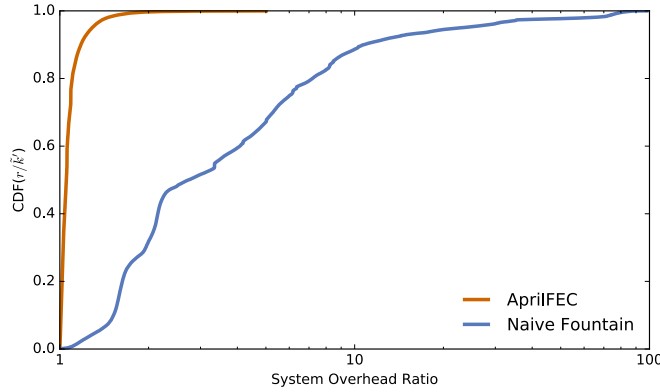

Fig. 6. CDF of the overhead added by the AprilFEC and naive fountain protocols. The overhead levels added by the systems are not comparable, with AprilFEC averaging only 8 percent overhead compared to the naive fountain's 646 percent. This prohibitive overhead level makes the naive fountain approach impractical for real-world applications.

if a more efficient fountain code were available (see discussion of code selection in Sec. III).

*2) System Overhead:* The system overhead ratio compares the number of encoded fragments actually received, $r$, to the number of fragments needed for decoding, $\tilde{k}'$. A protocol could achieve a system overhead ratio of 1.0 only if it perfectly estimates the number of fragments it needs to transmit.

Fig. 6 shows a CDF of the system overhead ratio for AprilFEC and the naive fountain. This plot illustrates the impracticality of the naive fountain approach for most applications. Because it makes no attempt to estimate the number of fragments to transmit, the naive fountain is entirely reliant on ACKs sent by the receiver to cut off its transmission. These ACKs are always subject to some delay and may be lost in transmission, in which case the transmission continues until a time limit is reached. This leads to an average overhead

level of 646 percent. In contrast, AprilFEC adds very little system overhead, averaging only 8 percent. The $1^{st}$ and $99^{th}$ percentiles are 0 percent and 60 percent, respectively.

The code and system overhead of AprilFEC combine for an average overall overhead of about 32 percent. The $1^{st}$ and $99^{th}$ percentiles of the combined overhead are 19 percent and 94 percent, respectively.

## VI. CONCLUSION

In this paper, we introduce AprilFEC, a system designed to mitigate the significant and unpredictable packet loss that occurs in robotic networks. AprilFEC adaptively applies forward error correction to input data to increase the likelihood of successful transmission while minimizing added overhead. AprilFEC enables reliable transport of large quantities of data across lossy, time-varying wireless links. We have shown that AprilFEC outperforms TCP by as much as 54 percent at the task of periodically delivering image files during simulated multi-robot trials while adding nominal amounts of overhead to the network.

## ACKNOWLEDGMENTS

This material is based upon work supported by the National Science Foundation Graduate Research Fellowship Program under Grant No. DGE 1256260. Any opinions, findings, and conclusions or recommendations expressed in this material are those of the authors and do not necessarily reflect the views of the National Science Foundation.

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
