# OpenReview forum: "AprilFEC: Real-Time Channel Estimation and Adaptive Forward Error Correction"
_roboticsfoundation.org/RSS/2017/RCW_Workshop/-_Proceedings_

### Review · AnonReviewer3 · 2017-06-27
**Overall, the paper is sound and the idea of combining fountain codes with kernel density estimation seems promising.**

**Rating:** 4
**Confidence:** 2

**Review:**

This paper proposes a transport protocol called AprilFEC, targeting links of a network with non-stationary varying packet loss rate distributions. The key insight is to use fountain codes, an encoding which split a message in a manner that enables redundancy and recoverability of the original message with high probability.

The algorithm uses kernel density estimation of the transmission loss rate, uses the estimated PDF in combination with probabilistic guarantees of the underlying fountain code to calculate the number of encoded fragments to send. Encoded messages are then sent (with time-varying fragment cardinality) until reception of an ACK (acknowledgment) message or timeout. The authors claim the resulting approach is more reliable than TCP and UDP and has a lower overhead than naïve usage of fountain codes for transmission. The approach is empirically evaluated against several approaches on a set of data-driven local network emulator.

Overall, the paper is sound and the idea of combining fountain codes with kernel density estimation seems promising. The experiments show reasonable improvements using their methods. Some suggestions and questions the authors should try to address in their revision/future work:

-	How does the approach compare against Marcotte and Olson’s method (citation [7] in the paper)? Both approaches use FEC, estimate packet loss rates, and adaptively change transmission characteristics per the estimates. More detailed discussions or (especially) empirical comparisons would be very useful.

-	Any comparisons or discussions of latency using the various methods tested? For example, this seems important when comparing against TCP (i.e., can you not get 100% reliability from TCP if you increase wait time?)

-	Since loss rate PDF estimation is claimed as part of the contribution, it would be useful to see some plots of its accuracy using the real-world data. Some discussion on how to tune the kernel estimation hyperparameters would be useful to readers.

-	The authors should provide a discussion of major limitations of their approach.

-	In Fig. 4, why does the max possible transmission events tend to increase with higher packet loss rates?

-	Include error bars on all subplots where applicable (e.g., Fig. 4).

-	In Alg. 1, second line after the first For Loop, I believe $p$ should be $p_t$.

---

### Review · AnonReviewer1 · 2017-06-28
**Paper introduces AprilFEC, a novel network protocol that uses erasure codes to intelligently encode data transmitted based on packet loss rates to for improved data reception in robot networks.**

**Rating:** 4
**Confidence:** 2

**Review:**

+ Idea of using erasure codes is interesting for efficient wireless transmission between robots
+ Demonstrates improved transmissions with AprilFEC in comparison to vanilla TCP/UDP transmissions

- The authors seem to confuse transport mechanisms such as TCP/UDP and application layer encoding such as the ones proposed. It is likely possible to get similar gains with erasure codes being used over TCP. Typically, the switch from TCP to UDP is in applications that require low latency where the overhead of SYN-ACK handshake done by TCP is too much (video streaming) or very short communication where the handshake overhead is in the order of data being transmitted (QUIC)
- The problem of transmission in robot networks is a large one. This work is specific to lossy links. Further, this algorithm works reasonably well if the losses are predictable temporally. However, wireless losses can be dependent on other factors such as occlusions, other users in the AP range, and attenuations for other reasons. In these cases, such a system might not work very well
- Experimental results are preliminary. It would be interesting to expand this study to involve multiple robots running an actual application so the trace-driven testing can mimic real applications
- There has been a lot of work on similar lines in the field of sensor networks, mesh networks and other wireless systems. This includes the large body of work in delay-tolerant networks as well as mobile adhoc networks. In order to distinguish this effort, I would recommend applying such a network protocol to a specific multi-robot application